# Longer Leukocytes Telomere Length Predicts a Significant Survival Advantage in the Elderly TRELONG Cohort, with Short Physical Performance Battery Score and Years of Education as Main Determinants for Telomere Elongation

**DOI:** 10.3390/jcm10163700

**Published:** 2021-08-20

**Authors:** Sofia Pavanello, Manuela Campisi, Alberto Grassi, Giuseppe Mastrangelo, Elisabetta Durante, Arianna Veronesi, Maurizio Gallucci

**Affiliations:** 1Section of Occupational Medicine, Department of Cardiac, Thoracic, Vascular Sciences & Public Health, University of Padova, 35128 Padova, Italy; manuela.campisi@unipd.it (M.C.); giuseppe.mastrangelo@unipd.it (G.M.); 2Unit of Occupational Medicine, University Hospital of Padova, 35128 Padova, Italy; 3Department of Statistical Sciences, University of Padua, 35121 Padova, Italy; alberto.grassi@studenti.unipd.it; 4Immunohematology and Transfusional Medicine Service, Local Health Authority n. 2 Marca Trevigiana, 31100 Treviso, Italy; elisabetta.durante@aulss2.veneto.it (E.D.); arianna.veronesi@aulss2.veneto.it (A.V.); 5Cognitive Impairment Center, Local Health Authority n. 2 Marca Trevigiana, 31100 Treviso, Italy; maurizio.gallucci@aulss2.veneto.it; 6Foundation for Interdisciplinary Geriatric Research (FORGEI), 31100 Treviso, Italy

**Keywords:** aging, lifespan, mortality deceleration, telomere length, longitudinal study, longevity, healthy aging

## Abstract

Leukocyte telomere length (LTL) represents a key integrating component of the cumulative effects of environmental, lifestyle, and genetic factors. A question, however, remains on whether LTL can be considered predictive for a longer and healthier life. Within the elderly prospective TRELONG cohort (*n* = 612), we aimed to investigate LTL as a predictor of longevity and identify the main determinants of LTL among many different factors (physiological and lifestyle characteristics, physical performance and frailty measures, chronic diseases, biochemical measurements and apolipoprotein E genotyping). We found an ever-increasing relationship between LTL quartiles and survival. Hazard ratio analysis showed that for each unit increase in LTL and Short Physical Performance Battery (SPPB) scores, the mortality risk was reduced by 22.41% and 8.78%, respectively. Conversely, male gender, Charlson Comorbidity Index, and age threatened survival, with mortality risk growing by 74.99%, 16.57% and 8.5%, respectively. Determinants of LTL elongation were SPPB scores (OR = 1.1542; *p* = 0.0066) and years of education (OR = 1.0958; *p* = 0.0065), while male gender (OR = 0.4388; *p* =  0.0143) and increased Disease Count Index (OR = 0.6912; *p*  =  0.0066) were determinants of LTL attrition. Longer LTL predicts a significant survival advantage in elderly people. By identifying determinants of LTL elongation, we provided additional knowledge that could offer a potential translation into prevention strategies.

## 1. Introduction

The aging of the population, also named the “gray” tsunami, is a growing social and public health issue. The impact of the phenomenon is forecasted by worrying data in Italy, one of the most elderly nations in the world [1]. In particular, the Northeast Veneto region area of Treviso is characterized by the highest longevity in Italy, especially among women [2]. The reasons are scantily known.

Aging is a peculiar and manifold process characterized by a gradual deterioration of the body’s capability to react to internal and/or external stress factors, resulting in an increased risk of illness and death [3]. This decline is the key risk factor of many human diseases, including cancer, diabetes, cardiovascular and neurodegenerative disorders. Not all people become eldery in the same manner [4]. Consequently, chronological age is not a consistent marker of the physiological decay of an individual [5], while biological age is expected to reflect the continuing changes within a person [6].

Accumulating evidence has revealed that leukocyte telomere length (LTL) reduction can be regarded as the early pillar of biological aging and may be the origin of cellular dysfunction [3]. Telomeres are short sequences of DNA repeated (TTAGGG) in association with various proteins forming a complex that is essential in preserving the genome stability of the cells [7]. In normal somatic cells, telomeric repeats are reduced by 30–200 bp after each mitotic division. However, telomeres, such as triple G-containing sequences, represent a susceptible target for genotoxic compounds, oxidative stress, and inflammation damages that directly accelerate telomere shortening. In our previous works, we showed that LTL erosion is related to risk factors for age-related diseases, including obesity [8], oxidative stress [9] and inflammatory responses [10], exposure to lifestyle (smoke, alcohol) [11], occupational pollutants [12] and cancer [13]. More recently, we found that in 585 residents in Northeast Italy, the everyday life exposure to polluted air, in particular indoors, loses LTL, particularly in males [14]. Therefore, the rate at which telomeres shorten provides a sturdy biomarker of aging and general health.

However, one of the most discussed questions in gerontology research is whether LTL is prognostic for a long and healthy life. The problem in interpreting the results of several studies arises from the heterogeneity of the factors that affect LTL, including genetics and environment, and in particular the availability of follow-up data both of LTL and survival of enrolled subjects.

The Treviso Longeva (TRELONG) study is a longitudinal cohort study initiated in 2003, which encompassed 668 seniors, women and men, over the age of 70 years, living in Treviso, the town with the highest longevity in North-East Italy [2]. The Trelong Study has investigated different topics such as frailty, cognitive impairment, Mediterranean diet and genetic aspects [15,16,17,18,19,20].

In view of the above, we investigated LTL within the prospective TRELONG cohort with two main aims: to ascertain whether this biomarker is predictive of longevity and to identify the main determinants in LTL among the personal characteristics (age, sex, marital status, education, occupation, etc.), physiological and lifestyle features, physical performance and frailty measures, chronic diseases, biochemical measurements and apolipoprotein E (APOE) genotyping.

## 2. Materials and Methods

### 2.1. Study Population

A complete description of the TRELONG study has been previously reported [15]. Eligible participants lived in the municipality of Treviso in North-East Italy. At the beginning of this study in 2003, Treviso had a resident population of 81,700. After the selection of the 13,861 inhabitants over 70 years of age (17% of total inhabitants) from the residents listed in the Registry Office of Treviso, participants were systematically sampled, planning to include 100 participants according to gender and 10-year-age group, 125 women and 125 men in the 70–79-year-age group, and all people over 100 years. A total of 668 people were selected: 311 men and 357 women (mean age 84.0 ± 8.0 years, range 70.0–105.5 years) [15]. Seven- and ten-year follow-ups were then performed. Using an interview, we collected data assessing clinical, lifestyle and demographic information, along with measures of physical performance and frailty, and a blood sample (30 mL). All were performed at home in three time points: in 2003 (baseline wave 1), in 2010 (wave 2) and 2013 (wave 3) [15]. Survival data were collected annually since 2003. In the present work, a total of 613 subjects for whom a sufficient DNA aliquot was still available for the telomere analysis in at least one of the three waves (2003, 2010 and 2013) were studied. All three measurements of LTL (at wave 1, wave 2, and wave 3) were available for only 162 people, because of mortality, impracticability of venous sampling or, rarely, refusal. Ours is, therefore, an observational cohort prospective study. The study protocol was approved by the Ethical Committee of the National Institute on Research and Care of the Elderly (INRCA, Ancona, Italy, prot. n. 50/02 on 22 March 2002). All participants and/or their caregivers provided a written consent.

### 2.2. Demographic, Physiological and Lifestyle Characteristics

Gender, age at the time of baseline in 2003, marital status, education in years, institutionalization, Body Mass Index (BMI), hypovisus, hearing loss, smoking status and alcohol intake were detected through home interview. To assess physical activity, participants were asked whether they took walks and/or did gardening every day. Data on eating habits were collected in 2003 (baseline) by questioning the participants about the regular intake (yes/no) of a list of foods. The statistical analyses showed that answers were substantially the same for the whole sample, suggesting all participants had the same eating habits [15].

### 2.3. Assessment of Physical Performance and Frailty Measures

The Short Physical Performance Battery (SPPB) was used to measure frailty via standing balance, walking and chair stand tests [21]. Standing balance tests included tandem, semi tandem and side-by-side stands. To test walking speed, each participant was timed for two walks on an 8-foot walking course. The faster of the two walks was used. The chair stand test assesses the ability to rise from a chair. Participants were asked to stand up from a chair one time. If successful, they were further requested to stand up and sit down five times as quickly as possible and were timed from the initial sitting position to the final standing position at the end of the fifth stand. Performance categories were created for each set of performance measures to permit analyses that included those unable to perform a task. For the 8-foot walk and repeated chair stands, those who could not complete the task were assigned a score of 0. Those completing the task were assigned scores of 1–4 corresponding to the quartiles of time needed to complete the task, with the fastest times scored as 4. The three tests of standing balance were considered as hierarchical in difficulty and assigned a single score of 0–4 for standing balance. A summary performance scale was created by adding up the category scores for walking, chair stand and balance tests [21]. The physical frailty index encompassed the chair stand and walking speed tests [22]. In particular, a severe frailty state was defined when the subject was unable to stand up from a chair and showed a walking speed slower than 0.6 m/s (2.16 km/h). Moderate frailty-state was when the subject showed impairment in one of the two tests. A robust (non-frail) state was defined when the subject was able to stand up from a chair without the use of the arms and showed a walking speed faster than 0.6 m/s.

Physical strength was also measured via handgrip using a Collins dynamometer (Witte GMBH Chirurgische Instrumente, Adult Size, 12.7 × 5.7 cm, Solingen, Germany), lifting a 2 kg weight over the head and taking the average of three assessments with the stronger hand [17]. Physical frailty was also measured through carrying out other exercises with the hands such as lifting a weight of 2 kg above the head using both arms and also bringing both hands behind the head and behind the shoulders [15]. Weight lifting was classified as: (1) not lifted; (2) lifted to the height of the shoulder; (3) lifted to the height of the head; or (4) lifted over the head. Disability was measured using the Activities of daily living (ADL) and Instrumental activities of daily living (IADL) score according to Lawton and Brody (1969) [23].

### 2.4. Assessment of Chronic Disease and Other Covariates

In-person interviews and examination of medical records provided information on the history of chronic diseases such as cerebrovascular diseases, chronic renal insufficiency, tumors and others (Table 1). Using this information, a Charlson comorbidity index (CCI) [24] as well as a Disease count index (DCI) [25] were determined. DCI was obtained by summing up the number of coexisting diseases for each subject.

The presence of cognitive decline was measured globally using the Mini-Mental State Examination (MMSE), range 0/30 [18]. Raw scores were corrected for age and education. Depression was assessed with the Geriatric Depression Scale (GDS) [18].

### 2.5. Biochemical Measurements

A fasted morning blood was collected at home from each participant and transported to the Clinical Chemistry Laboratory of Treviso Hospital, within 30 min before 9.00 a.m. (12h fasting) as previously described [15]. Briefly, to evaluate the general health status of participants, routine hematological and clinical chemistry tests were performed immediately using standard laboratory methods. Serum, plasma (tubes containing anticoagulant Na-citrate) and whole blood (tubes containing anticoagulant K2 EDTA) were prepared, and blood leukocytes were isolated to be stored in the biobank of the Transfusional Center Unit.

Serum biochemical markers, such as albumin, total and high-density lipoprotein (HDL)-Cholesterol, creatinine, etc., were determined by commercial methods (Abbot Laboratories, Abbot Park, IL, USA) on an Aeroseth Analyzer. Hematologic indices were assessed with the ADVIA 120 Hematologic System (Siemens Healthcare Diagnostics, Deerfield, IL, USA). Coagulation tests were performed using a plasma-citrate CA-7000 (Sysmex Corporation, Kobe, Japan) and erythrocyte sedimentation rate (ESR) using an automatic analyzer Test 1 TH (Alifax S.p.A, Polverara, Italy). Immunoglobulins and high sensitivity C-reactive protein (hsCRP) in serum were measured with Behring Nephelometer Analyzer II (BNA) (SimensHealtcare Diagnostics, Deerfield, IL, USA). Plasma interleukin-6 (IL-6) [19] was quantified using ELISA (UltraSensitive ELISA kit, BioSource, San Diego, CA, USA). The minimum detectable amount was 0.10 pg/mL. Plasma insulin-like growth factor-1 (IGF-1) was quantified using ELISA (Nonextraction IGF-1 ELISA kit, Diagnostic Systems Laboratories, Webster, TX, USA), with a minimum detectable amount of 10 ng/mL.

### 2.6. DNA Extraction from Mononuclear Cells (PBMC)

Fasting peripheral blood samples (30 mL) were collected by venipuncture; one aliquot was used to separate mononuclear cells (PBMC) by a standard Ficoll centrifugation procedure as previously described [15,19]. PBMC pellets were washed with ice-cold PBS, divided into aliquots and stored at −80 °C for further analysis. Genomic DNA was extracted from PBMC pellet using a vacuum-based semi-automated nucleic acid extractor (AB6100, Applied Biosystems, Foster City, CA, USA), checked for concentration by a UV-spectrophotometer (Eppendorf, Hamburg, Germany) and stored at −20 °C. Samples of genomic DNA available derived from the three waves were sent to the Laboratory of Environmental Mutagenesis and Genomic—Occupational Medicine, Department of Cardiac, Thoracic, and Vascular Sciences and Public Health, University Hospital of Padua, for the analysis of leukocyte telomere length (LTL), with a file reporting all information regarding qualitative (260/280 nm) and quantitative (ng/μL) description and volume of each sample (μL).

### 2.7. APOE Genotyping

APOE genotype was determined by PCR using the following primers: for 5_-tcggccgcagggcgctgatgg-3_; rev 5_-ctcgcgggcccccggccccggcctggta-3_; the resulting 332-bp PCR product was digested with CfoI (Roche, Basel, Switzerland) as previously described [26]. Briefly, the APOE genotype was then assessed after loading the corresponding enzymatic digestions on a capillary electrophoresis unit alongside with standard reference marker to allow the instrumental lining-up (Agilent Technologies, Santa Clara, CA, USA).

### 2.8. Leukocyte Telomere Length (LTL) Analysis

LTL in genomic DNA was appraised by the real-time quantitative polymerase chain reaction (PCR) as previously described [14,27] according to the reference method [28]. In brief, this method measures the relative LTL by estimating in genomic DNA the proportion of telomere repeat copy number (T) in relation to a single copy gene (S) (T/S ratio). T/S ratio is therefore the unit of measurement of LTL. The single-copy gene employed in this investigation was the human β-globin (hbg) [27]. A “six-point” reference curve was built from a serially diluted DNA pool, varying from 15 to 0.47 ng in each plate, in order to calculate the relative quantities of T and S in ng of samples to be examined. The DNA pool was realized by taking an aliquot of genomic DNA from samples at random designated and derived from the three waves: baseline collection (T0) and consecutive collection after a 7-years (T1) and 10-years (T2) follow up. All samples and standards were analyzed in triplicate and the average of the 3 T/S ratio measurements was considered in the statistical analyses. In brief, Qiagility (QIAGEN, Milano, Italy), which enables a high-precision PCR set up, was used for transferring 10 μL of reaction mix and 5 μL of DNA (5 ng/μL) in a 96-well plate. All PCR reactions were performed on a SteponePlus Real-Time PCR System (Applied Biosystems, Monza, Italy). A primer pair for a beta-globin single copy gene (hbgu and hbgd) [14], as well as a telomere primer pair (telg and telc) as described in Cawthon [28], were used in the reaction mix. The thermal cycling profile for both amplicons began with incubation at 95 °C for 10 min to activate the AmpliTaq DNA polymerase. For telomere PCR, activation was followed by 2 cycles of 15 s at 95 °C, 15 s at 49 °C, and 35 cycles of 15 s at 95 °C, which was then followed by 10 s at 62 °C, and 15 s at 74 °C. For hbg, activation was followed by 35 cycles of 15 s at 95 °C and 1 min at 58 °C. LTL was analyzed in triplicate for all samples of genomic DNA available and replicated on different days. The average of the three T measurements was divided by the average of the three S measurements to calculate the average T/S ratio, i.e., the relative telomere length. A measure was considered acceptable if the standard deviation among triplicate measures was <0.25. The coefficient of variation for the average T/S ratio of samples analyzed over three consecutive days was <9%, which was similar to the reproducibility originally reported for this method.

### 2.9. Statistical Analysis

The distribution of LTL at the first wave (2003) according to characteristics of the study population was appraised with Mann–Whitney U Test and Kruskal–Wallis test.

#### 2.9.1. Dependence of LTL on Age

Considering the subgroup of *n* = 162 subjects with available three telomere measurements (in 2003, 2010, 2013), we used the Friedman test to compare the longitudinal distribution of the LTL at the three waves.

Since LTL could be measured more than once on the same person (repeated measures taken over time) to account for both within-person and across-person variability, we used the linear mixed model to appraise the influence of chronological age and gender (independent variables) on the logarithmically-transformed LTL (dependent variable), since the telomere distribution was non-Gaussian. The addition of the random effects for each subject was useful to take into account the individual heterogeneity and the dependence between the measurements made on the same subject.

#### 2.9.2. Survival Analysis: Kaplan–Meier Curves

Survival analysis was performed using the non-parametric Kaplan–Meier curves analysis on all 613 subjects. The survival time was calculated for each person starting from the date of the interview of the year corresponding to the first LTL measurement available until the date of death or end of follow-up (25 June 2019). The survival curves were estimated stratifying the subjects in two groups based on the median LTL, or in four groups based on the quartiles (1° quartile: 0.46 ≤ LTL < 1.57; 2° quartile: 1.57 ≤ LTL < 2.11; 3° quartile: 2.11 ≤ LTL < 2.86; 4° quartile: 2.86 ≤ LTL < 7.44). The difference between survival curves was estimated by the log-rank test and the Peto–Peto log-rank test, which is preferable when there are intersections among the survival curves.

#### 2.9.3. Survival Analysis: Cox Proportional Hazards Model

To evaluate the effect of LTL on survival adjusting for other factors, a semi-parametric Cox proportional hazards model was constructed using the same sample of 613 subjects. Before, the adequacy of the proportionality assumption of risks was assessed by analyzing the Schoenfeld residues. The analysis of the Schoenfeld residuals gave a non-significant result (*p* = 0.92) for each covariate and for the model as a whole; therefore, the hypothesis of proportionality of the risks was accepted. Based on the relevant literature, gender, age, CCI and SPPB were used as covariates besides LTL in the Cox regression model. For the interpretation of the estimates, the hazard ratios of each variable were reported in the tables. Since LTL varied over time and there was more than one measurement for some subjects over the three waves, we used an extension of the Cox model that allows for the inclusion of time-dependent variables by the technique of episode splitting.

This model was further improved considering for all 613 subjects a statistical weight computed so that the TRELONG study population was representative per gender and age of the Italian population over seventy.

#### 2.9.4. Characterization of Subjects with Elongated Telomeres

To find the factors associated with an increase in LTL in all 613 subjects, we created a dummy variable, which was 1 if the subject had an increase in LTL between the first (2003) and the second (2010) waves or between the second (2010) and the third (2013) waves and 0 in absence of elongation. This was the outcome variable of a logistic regression model where the predictors were the variables reported in Table 1. A stepwise approach based on Rao’s score tests was applied to eliminate non-significant variables from the model.

Results were considered significant when a *p*-Value < 0.05 was obtained. The analyses were performed using the statistical software R version 3.6.3 [29].

## 3. Results

### 3.1. Descriptive Results

#### 3.1.1. Characteristic of Population at the Baseline and LTL Distribution

Table 1 reports the main TRELONG variables/characteristics at the time of baseline in 2003 (clinical, lifestyle and demographic information acquired through questionnaires, physical examination, and laboratory tests).

The distribution of LTL (2003) according to the characteristics of the study population is reported in Table 2. LTL decreased (*p* < 0.05) with age, institutionalization, hearing loss, moderate and severe physical frailty and presence of chronic diseases (DCI and CCI). LTL increased (*p* < 0.05) with increasing physical activity, better cognitive profile (MMSE score > 26), greater functional autonomy (ADL score = 6 and IADL score = 8), better physical performance (SPPB score ≥ 6 and effective placement of one’s hands behind the shoulders) and physical strength (hand gripscore ≥ 12.2 kg and full lifting of the weight of 2 kg above head). Furthermore, LTL increased with blood chemistry variables such as albumin, hemoglobin and insulin-like growth factor-1 (IGF-1).

#### 3.1.2. LTL and Age in the Longitudinal Study

Figure 1 shows the distribution of LTL in the subgroup of 162 subjects in the three waves. A reduction in both mean and variance is observed, indicating a reduction in telomeres with increasing age. Friedman’s test in fact leads to the rejection of the null hypothesis of equality on the average in the three groups (*p* < 0.001).

Table 3 shows the results of the mixed linear model on the logarithmically transformed LTL. The gender variable was removed from the model because it was not significant (*p* = 0.594). The estimated model indicates that for each additional year of age, LTL dropped by 7.96% (*p* < 0.001). The average trend estimated by the model on the original scale is shown in Figure 2. The residuals of the estimated model satisfy the assumption of normality according to the Shapiro–Wilk test (*p* = 0.064).

#### 3.1.3. LTL and Survival: Kaplan–Meier Curves

Figure 3 shows that the group with longer LTL (T/S > 2.11; curve in blue) had a consistently higher survival than the group with shorter LTL (T/S ≤ 2.11; curve in green). The median survival time, (corresponding to the survival time of 50% of subjects in each group), was 7.57 years for subjects with telomeres longer than 2.11 (T/S) and only 5.41 years for subjects with telomeres equal or shorter than 2.11 (T/S). The two adjacent 95% confidence interval limits did not overlap for most of the curve. The difference between the two survival curves was significant using both the log-rank test (*p* < 0.001) and the Peto–Peto log-rank test (*p* < 0.001), which was better because a slight intersection between the two survival curves was observed in the first period (see Figure 3).

Figure 4 shows that the median survival time was 4.67, 5.87, 6.73 and 8.79 years, respectively, for LTL quartiles 1 to 4 (1 ≥ 0.46 and <1.57; 2 ≥ 1.57 and <2.11; 3 ≥ 2.11 and <2.86; 4 ≥ 2.86 and <7.44). The Peto–Peto log-rank test rejected the hypothesis of equality between the survival curves in the four groups (*p* <0.001). The median survival time was over 4 years longer in the group with the longest telomeres (in purple in Figure 4) than in the group with the shortest telomeres (in red in Figure 4). In conclusion, there was an ever-increasing dependence between LTL and estimated survival: the longer the LTL, the longer the survival and therefore the lower the estimated mortality risk.

### 3.2. Outcome Results

#### 3.2.1. Determinants of Survival

Table 4 shows the results of the semi-parametric Cox proportional hazards model, estimated with the episode splitting technique. It can be seen that LTL had a significant effect on survival even in a multivariate analysis taking into account the effect of other factors such as age, gender, the CCI and the SPPB. In fact, all the variables included in the model (age, LTL, gender, CCI, SPPB) were very significant (*p* < 0.001). The coefficient associated with LTL and SPPB had a negative sign, indicating a protective effect of these variables with respect to mortality risk, while the sign was positive for other variables (age, gender (male), CCI), indicating an increase in risk. The interpretation of the hazard ratios for each variable can be the following: (1) age, for each additional year of the subject, there is an increase of 7.75% in the risk of mortality, all other factors being equal; (2) LTL, for each unit increase in the LTL, the risk of mortality decreases by 16.99% with the other explanatory variables being equal, age included; (3) Charlson index score, for each unit increase in the index, the risk of mortality increases by 9.65%, all other factors being equal; (4) SPPB index score, for each unit increase of the index, the mortality risk, all other factors being equal, is reduced by 10.01%; (5) gender, males have a 73.16% higher risk of mortality, all other factors being equal, compared to females.

Table 5 shows that after re-weighting the sample, the significant effects of all the explanatory variables did persist and the interpretation remained substantially the same. However, in this second model, the protective effect of LTL was slightly stronger (22.41% decrease per unit increase in the risk of mortality), while the protective effect of SPPB was slightly weaker (8.78% decrease per unit increase). Risk factors such as age (8.5% increase for each additional year), gender (74.99% higher risk for males) and CCI (16.57% growth per unit increase) all appeared to have a stronger effect when compared with those obtained from the model on the unweighted sample (Table 4).

#### 3.2.2. Determinants of LTL Elongation

Out of 613 subjects, overall 73 subjects (11.9%) presented an increase in LTL: they were 38 in the first period (2003–2010) and 35 in the second period (2010–2013). No subject registered an increase during the whole observation period (2003–2013).

Table 6 reports the results of fitting of logistic regression model based on Rao’s score test. Male gender was a risk factor for telomere attrition (OR = 0.4388; *p* = 0.0143) along with increase in the DCI values (OR = 0.6912; *p* = 0.0066). By contrast, SPPB scores (OR = 1.1542; *p* = 0.0066) and years of education (OR = 1.0958; *p* = 0.0065) were determinant of telomere elongation.

## 4. Discussion

### 4.1. LTL and Survival

A key finding from our study is the steadily increasing relationship between LTL and survival: the longer the LTL, the greater the survival and, therefore, a lower mortality risk is estimated. In particular, as shown in Figure 3, the group with the longest quartile LTL (T/S >2.86) had 8.79 median years of survival, which is over 4 years higher than that of the group with the shortest quartile LTL (<1.57 T/S). Furthermore, in a multivariate analysis taking into account the effect of other factors, LTL increased survival and therefore decreased mortality. In detail, for each unit increase in the LTL, the risk of mortality decreased by 22.41%, with the other explanatory variables being equal, age included (Table 5).

LTL has often been related to life expectancy, but contradictory results have been reported in older adults. Whereas some studies have reported that LTL attrition is a marker of mortality [30,31,32,33], others have not found such a relationship [34,35,36,37], making the relationship between LTL and survival in humans uncertain. Those authors that did not find this association mentioned variation in methods of LTL measurements [35] and variations among investigated populations, including a short age range (>75 years) [30,35,37] and diseases examined, where the elongation of telomeres could be a characteristic of the disease [36].

### 4.2. Determinants of Survival

Key survival determinants were not only LTL but also SPPB: for each unit increase in the index, the mortality risk was reduced by 8.78%, considering all other factors being equal, age included (Table 5). It is known that physical activity is correlated with reduced risks of all cause and cause-specific mortality [38]. Furthermore, a prospective cohort study showed that middle-aged and older adults achieved considerable longevity advantages by becoming more physically active, regardless of past physical activity [39]. In our study, we used the SPPB that may be more likely to capture the real health status of older persons. This test, with three different evaluations (walking speed, chair stand and balance time) [21,40], was thought an extremely sensitive indicator of global health status or vulnerability, revealing numerous underlying physiological disorders [41]. Our results are in line with one of the most complete meta-analyses with a proper sample size that definitively discovered the association between SPPB score and all-cause mortality [42].

On the other hand, the male gender, CCI and age were factors that threatened the survival, with a risk of mortality growing by 74.99% for males, by 16.57% per unit increase in CCI and by 8.5% for each additional year of age (Table 5). These data confirmed the current literature on gender inequity: men achieve higher mortality rates than women [43]. Human females’ longevity benefit may derive from the hormonal effect on inflammatory and immunological reactions [44]. CCI is a validated, simple and easily applicable method for appraising the risk of death and has been broadly applied as a predictor of long-term prognosis and survival [24,45]. In our study, CCI score gave an HR of 1.2 (Table 5) similar to that reported by Fraccaro et al. [46]; i.e., 1.3 HR in an adult cohort study from Salford (UK) from 2005 to 2014.

### 4.3. LTL in the Longitudinal Study

LTL distribution at the baseline in 2003 showed that LTL attrition was related to age (Table 2). Another key finding, by analyzing LTL at the three waves in the subgroup of 162 subjects, was LTL drop by 7.96% for each additional year of age (Table 3). The rate of LTL shortening we found was similar to that observed in the longitudinal population-based Bruneck Study (age 45–85 years), in which the telomeres lost on average 455 bp in length after 10 years, corresponding to an average loss of 45.5 bp per year that is equal to about 10% [47]. An overall lesser decrease (13.6% after 7 years follow-up) than our study was, however, found in a cohort study of 516 subjects aged 65–106 from Southern Italy [48]. This could be attributed to the major number of centenarians in whose LTL may be expected to increase after 90 years [49]. Lastly, our LTL drop is comparable to the 7.6% decrease in LTL reported by McCracken et al. [50] associated with an annual increase in air pollution. It could be speculated that our decrease in LTL may be attributed to the air pollution of the Veneto region, being part of the Pianura Padana, one of the most polluted regions in Europe.

Another noteworthy key finding of our study was the paradoxical increase in LTL in almost 12% of the study population. This finding, with almost exactly the same percentage, has been previously reported [34]. Loss of telomeric DNA caused by deterioration or incomplete replication seems to be balanced by telomere elongation, which involves the telomerase, a DNA polymerase that synthesizes TTAGGG repeats, inducing the de novo synthesis of additional repeats. This specific reverse transcriptase complex counters the replication-associated telomere shortening with the addition of G-rich telomeric repeats to the chromosome ends, thus efficiently stabilizing TL [51]. That the gain in LTL is merely a consequence of a high degree of measurement error can be rejected by the accurate quality controls with measurements in triplicate of T/S ratios that resulted in a standard deviation < 0.25 and with a coefficient of variation (CV), on average, less than 9%, in samples analyzed during three successive days, which is comparable to the reproducibility formerly described for this method [28].

### 4.4. Determinants of LTL Elongation

Among determinants eliciting LTL elongation were SPPB and years of education, while being male and DCI were factors that contributed to reduced LTL. Until now, few studies have explored the association between LTL and SPPB, with conflicting results. Pereira et al. [52], in a study with a small number of subjects, reported unexpectedly that shorter LTL was correlated with better physical capability evaluated by SPPB score. No association was found in a cohort of 136 patients aged 45–85 years with knee osteoarthritis pain, in which the SPPB test is considered a clinical measure for assessing physical function [53]. We can assert that, in general, in observational studies, greater physical activity or exercise was associated with longer LTL in various populations, whereas in a small number of longitudinal studies in which exercise interventions were performed to investigate the potential effect on LTL, such a relationship was not fully established [54]. It has been reported that physical exercise, affecting the balance between oxidative stress and antioxidants and altering methylation patterns, buffers LTL shortening [54]. Furthermore, physical training increases telomerase activity in myocytes and circulating mononuclear cells in rats [55] and in human leukocytes [55], thus providing another possible and potentially testable pathway to clarify how physical activity protects LTL [56].

The positive relationship between LTL and years of education agrees with previous findings on a positive association between educational attainment and LTL [57,58]. The implication of this finding suggests that education confers health benefits by limiting biological aging. Education, in fact, is one of the leading determinants of overall social status over the course of a person’s lifetime and can reduce exposure to chronic stress and/or increase one’s capacity to effectively deal with potential threats or stressors [59].

Instead, the role of the male sex as a negative factor for telomere elongation is consistent with the majority of previous cross-sectional and longitudinal studies establishing that women have longer telomeres than men [60] and that the rate of LTL shortening is slower in women than men [60]. Several realistic biological reasons can be formulated to elucidate this, including the action of estrogens [61], which can promote the production of telomerase [62] and protect against damage caused by reactive oxygen species [63].

Convincing evidence in the literature suggests the association between shorter LTL and several age-related diseases (such as cardiovascular diseases, neurodegenerative diseases, type 2 diabetes mellitus, premature aging syndromes and cancers) [30,64,65]. Therefore, it is not surprising that the concomitant existence of a higher number of diseases, indicated by a higher DCI, is a risk factor that contributes to reduce LTL.

### 4.5. Strengths and Limitations

Whether LTL can be considered a predictor for longevity and healthy life is one of the most questioned issues in the field of gerontological research. Problems in results’ interpretation stem from the heterogeneity of factors influencing LTL, including genetics and environment, and in particular from the available follow-up data for both LTL and survival for recruited samples.

Our study was conducted in Treviso, in Veneto, a region of North-East Italy, which is characterized by the highest longevity, in particular among women [2]. Prevalence of the APOEε4 allele in Treviso is equal to 16.3%, which is lower than the 25% mostly observed for northern European populations without dementia [26], and this can explain, at least in part, the greater longevity of women in the Treviso area. It should be noted that the Northern population, including Venetians, represents a quite homogeneous population from the genetics point of view, which is different from other populations studied so far in Italy. In the ancestors of Northern Italians, climate-related selective pressures have influenced an adaptive evolution that make Northern Italian people less prone to develop accelerating aging disorders, including diabetes type 2 and obesity [66].

In our study, we determined LTL during ten-year follow up in the prospective cohort “Treviso Longeva (TRELONG)”, a very well-characterized study population. In particular, this is a systematic sample of subjects aged 70–106 years, drawn from the list of residents of the municipality of Treviso. Therefore, the study sample is highly representative of the community and of the real world [15]. Furthermore, the cohort is well characterized with several types of information, including physiological variables and lifestyles, physical performance and frailty measures, assessment of chronic disease and other covariates besides biochemical measurements and APOE genotyping [26].

We used the q-RT PCR technique to assess LTL in triplicate. This fast and highly sensitive method requires relatively small quantities of DNA and allows high throughput. Since an almost totally automated workflow from DNA extraction to LTL analysis was used, pre-analytical factors and analytical factors did not vary amongst analyses and the same pool DNA was used as a reference standard in all q-RT PCR reactions. In this way, the interbach variation was under control, minimizing measurement error and warranting a CV of less than 9%.

Moreover, in this study, the start date of the survey was the day of the interview rather than the date of birth so that the measurements relating to the lengths of the telomeres of the interviewees who had very different ages were comparable. In this case, then, the duration of the episode indicates the remaining survival time for the individual with that LTL.

Another strong point is that the Cox model estimate for time-dependent variables was further improved considering that for all 613 subjects a statistical weight was computed so that the TRELONG study population was representative per gender and age of the Italian population over seventy. Therefore, the results could be more easily extended to the entire population over seventy at the national level. It should be noted, however, that while the sample of the present study was found in a single city, a sample detected at the national level would be affected by the different geographical distribution of the subjects recruited.

## 5. Conclusions

Our results demonstrate that LTL could be a powerful tool for predicting survival in our elderly population. By identifying SPPB as a key survival determinant, our study suggests that the regular application of the SPPB in clinical practice may offer helpful prognostic information regarding the risk of all-cause mortality. This study also confirms that male gender, CCI, and, therefore chronological age, are threats for survival.

Furthermore, by identifying SPPB scores and years of education as positive and DCI as a negative determinant of LTL elongation, we recognized factors on which to converge to improve prevention strategies.

## Figures and Tables

**Figure 1 jcm-10-03700-f001:**
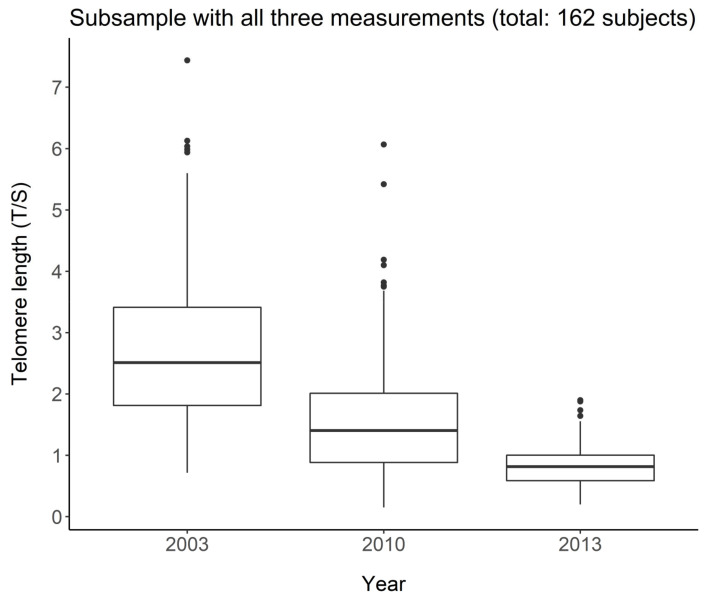
Telomere length change over the three waves in the group of 162 subjects with all three measurements.

**Figure 2 jcm-10-03700-f002:**
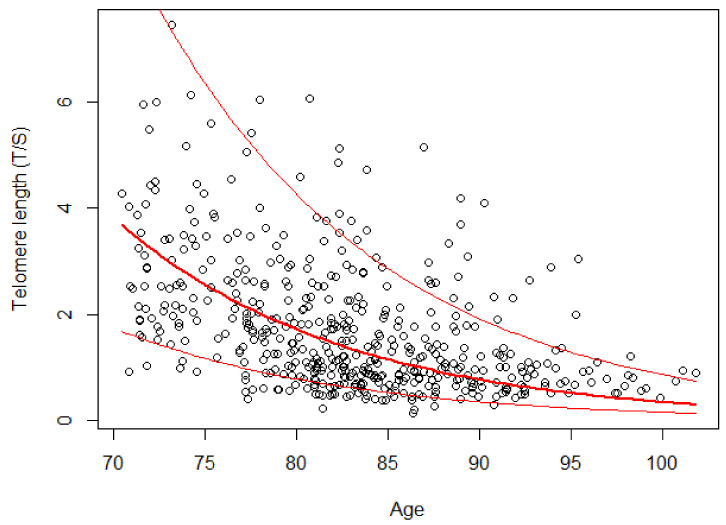
Telomere length versus age in the group of 162 subjects. The red lines indicate the random-effects model estimated on a logarithmic scale and the range of variation due to the addition of the individual random effects.

**Figure 3 jcm-10-03700-f003:**
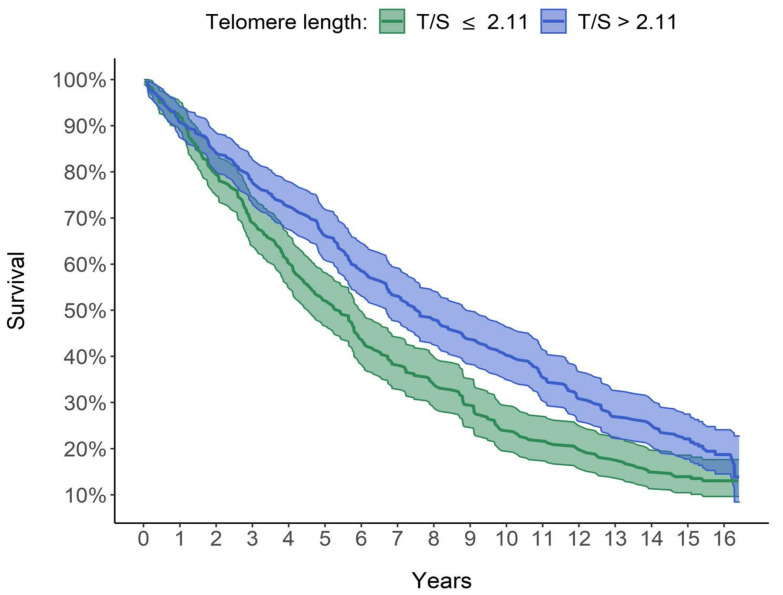
Graphic comparison of the survival of the 2 groups. Survival curves estimated using Kaplan–Meier method. The median survival time is 5.41 years for subjects with T/S ≤ 2.11 and 7.57 years for subjects with T/S > 2.11.

**Figure 4 jcm-10-03700-f004:**
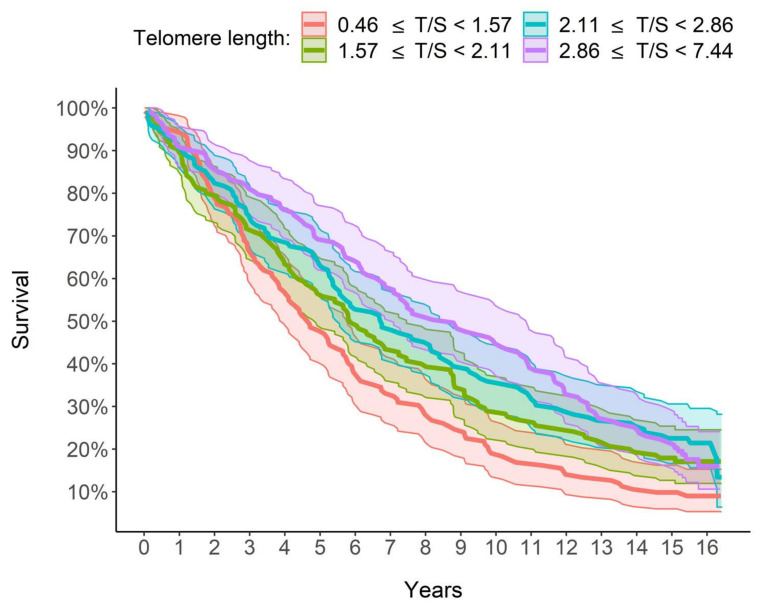
Graphic comparison of the survival of the 4 groups/quartiles. Survival curves were estimated using Kaplan–Meier method. From the group with the shortest LTL to the one with the longest LTL, the median survival time is, respectively, 4.67, 5.87, 6.73 and 8.79 years.

**Table 1 jcm-10-03700-t001:** Characteristic of TRELONG study population at the beginning of the study/at the first wave *n* = 613.

Variable		Mean ± Standard Deviation (or Number (%))	Number of Missing Data
Demographic, physiological variables and lifestyles			
Gender (*n* (%))	Female	331 (54.00%)	0
	Male	282 (46.00%)	
Age (years)		83.99 ± 8.06	0
Marital status (*n* (%))	Unmarried	60 (9.80%)	1
	Married	266 (43.46%)	
	Divorced	5 (0.82%)	
	Widower	281 (45.92)	
Education (years)		6.98 ± 4.82	3
Institutionalization (*n* (%))		86 (14.03%)	0
Body Mass Index (BMI) (kg/m^2^)		24.85 ± 4.11	50
Hypovisus (*n* (%))		207 (34.97%)	21
Hearing loss (*n* (%))		210 (37.17%)	48
Smoking (*n* (%))		247 (40.29%)	0
Wine intake * (*n* (%))		363 (59.22%)	0
Physical activity ** (*n* (%))		430 (70.15%)	0
Biochemical measurements			
Total cholesterol (mg/dL)		214.4 ± 43.57	59
Low-density lipoprotein (LDL) cholesterol (mg/dL)		135.93 ± 37.44	59
High-density lipoprotein (HDL) cholesterol (mg/dL)		55.99 ± 14.84	50
Erythrocyte sedimentation rate (ESR) (mm/h)		35.58 ± 21.44	50
Fibrinogen (mg/dL)		349.24 ± 75.94	52
Albumin (g/dL)		4.18 ± 0.36	59
Creatinine (mg/dL)		1.06 ± 0.48	49
Hemoglobin (g/dL)		13.64 ± 1.64	61
Platelets (*n* × 10^3^/microL)		237.3 ± 75.28	50
White cells (*n* × 10^3^/microL)		6.46 ± 1.7	50
Red cells (*n* × 10^6^/microL)		4.44 ± 0.53	50
High sensitivity C-reactive protein(hs-CRP) (mg/L)		0.61 ± 1.43	60
Interleukin-6 (IL-6) (pg/mL)		1.68 ± 2.09	94
Lg IL-6		−0.07 ± 1.09	94
Insulin-like growth factor-1 (IGF-1) (ng/mL)		178.74 ± 77.93	53
Assessment of chronic diseases and other covariates			
Hypertension (*n* (%))		368 (60.03%)	0
Sistolic blood pressure (mm/hg)		144.88 ± 20.09	29
Diastolic blood pressure (mm/hg)		81.22 ± 11.38	29
Disease Count Index (DCI)		2.22 ± 1.62	0
Charlson Comorbidity Index (CCI)		5.69 ± 2.01	0
Ischemic heart disease (*n* (%))		108 (17.62%)	0
Heart failure (*n* (%))		50 (8.16%)	0
Malignant neoplasm (*n* (%))		100 (16.31%)	0
Diabetes (*n* (%))		101 (16.48%)	0
Ictus (*n* (%))		59 (9.62%)	0
Chronic obstructive pulmonary disease (*n* (%))		31 (5.06%)	0
Mini Mental State Examination (MMSE)		23.89 ± 6.65	0
Geriatric depression scale (GDS)		3.86 ± 3.7	30
Physical performance and frailty measures			
Frailty (*n* (%))	Absent	131 (21.37%)	0
	Moderate	247 (40.29%)	
	Severe	235 (38.34%)	
Activities of daily living (ADL) score		5.03 ± 1.83	17
Instrumental activities of daily living (IADL)score		5.28 ± 3	26
Short physical performance battery (SPPB)		5.45 ± 3.78	0
Handgrip (kg)		12.17 ± 9.79	1
Lift 2 kg (*n* (%)) ***		394 (67.35%)	28
Hands behind the head (able) (*n* (%))		506 (83.91%)	10
Hands behind the shoulders (able) (*n* (%))		502 (83.25%)	10
LTL (T/S)		2.36 ± 1.14	41
APOE Genotyping		89 (15.92%)	54

* Subjects consuming at least 1 glass of wine/day; ** Subjects regularly walking or doing gardening activities; *** Percentage of those who manage to carry the weight over their heads.

**Table 2 jcm-10-03700-t002:** Distribution of telomere length according to characteristics of the study population.

Variables (Wave 2003)		LTL (T/S)	*p*-Value (Mann–Whitney Test or Kruskal–Wallis Test)
Demographic, physiological variables and lifestyles			
Gender	Male	2.35	0.9729
	Female	2.361	
Age (Medianyears)	<83	2.614	<0.0001
	≥83	2.109	
Marital status	Unmarried	2.304	0.6236
	Married	2.403	
	Divorced/Separated	2.716	
	Widower	2.308	
Education (years)	<5	2.37	0.8859
	≥5	2.357	
Institutionalization	No	2.41	0.0079
	Yes	2.045	
Body Mass Index (BMI) (kg/m^2^)	Underweight (<18.5)	1.993	0.2774
	Normal (18.5–24.9)	2.307	
	Overweight	2.512	
	(25.0–29.9)		
	Obese (30.0–39.8)	2.364	
Ipovisus	No	2.349	0.8921
	Yes	2.385	
Hearing loss	No	2.45	0.0002
	Yes	2.174	
Smoking	No	2.301	0.3118
	Yes	2.434	
Wine intake	0	2.342	0.6453
	2 glasses/day	2.351	
	+2 glasses/day	2.409	
Physical activity	Able (yes)	2.457	0.0016
	Unable (no)	2.13	
Biochemical measurements			
Total cholesterol (mg/dL)	<214	2.355	0.8047
	≥214	2.386	
Low-density lipoprotein (LDL) cholesterol (mg/dL)	<134	2.381	0.7521
	≥134	2.36	
High-density lipoprotein (HDL) cholesterol (mg/dL)	<54	2.288	0.507
	≥54	2.387	
Erythrocyte sedimentation rate (ESR) (mm/h)	<32	2.323	0.7566
	≥32	2.36	
Fibrinogen (mg/dL)	<339	2.362	0.9624
	≥339	2.329	
Albumin (g/dL)	<4.2	2.249	0.0236
	≥4.2	2.457	
Creatinine (mg/dL)	<1	2.343	0.8339
	≥1	2.341	
Hemoglobin (g/dL)	<13.8	2.292	0.0132
	≥13.8	2.44	
Platelets (*n* × 10^3^/microL)	<232	2.37	0.8637
	≥232	2.314	
White cells (*n* × 10^3^/microL)	<6.35	2.332	0.7527
	≥6.35	2.351	
Red cells (*n* × 10^6^/microL)	<4.49	2.289	0.1161
	≥4.49	2.394	
High sensitivity C-reactive protein (hs-CRP) (mg/L)	<0.21	2.393	0.8004
	≥0.21	2.351	
Interleukin-6 (IL-6) (pg/mL)	<0.9	2.395	0.4656
	≥0.9	2.374	
Insulin-like growth factor-1 (IGF-1) (ng/mL)	<167	2.268	0.0423
	≥167	2.442	
Assessment of chronic diseases and other covariates			
Hypertension	No	2.398	0.4552
	Yes	2.329	
Systolic blood pressure (mm/hg)	<144.6	2.393	0.6309
	≥144.6	2.346	
Diastolic blood pressure (mm/hg)	<80.5	2.337	0.3803
	≥80.5	2.402	
DiseaseCount Index (DCI)	0–2	2.415	0.0296
	3–10	2.262	
CharlsonComorbidity Index (CCI)	3–5	2.472	0.0091
	6–14	2.25	
Ischemicheartdisease	No	2.379	0.269
	Yes	2.252	
Heartfailure	No	2.373	0.4305
	Yes	2.172	
Malignant neoplasm	No	2.337	0.8135
	Yes	2.455	
Diabetes	No	2.328	0.3831
	Yes	2.494	
Ictus	No	2.372	0.3477
	Yes	2.218	
Chronic obstructive pulmonary disease	No	2.369	0.3647
	Yes	2.113	
Mini Mental State Examination (MMSE)	≤26	2.253	0.0119
	>26	2.454	
Geriatric depression scale (GDS)	≤3	2.409	0.272
	>3	2.323	
Physical performance and frailty measures			
Frailty	Absent	2.526	0.0015
	Moderate	2.455	
	Severe	2.173	
Activities of daily living (ADL) score	Able = 6	2.466	0.0014
	Unable ≤ 5	2.079	
Instrumental activities of daily living (IADL)score	Able = 8	2.611.	<0.0001
	Unable ≤ 8	2.2	
Short physical performance battery (SPPB)	<6	2.24	0.014
	≥6	2.466	
Handgrip (kg)	<12.2	2.21	0.0013
	≥12.2	2.524	
Lift 2 Kg	Able	2.462	0.0094
	Unable	2.216	
Hands behind the head	Able	2.398	0.0727
	Unable	2.158	
Hands behind the shoulders	Able	2.402	0.0477
	Unable	2.149	
APOE Genotyping	ApoE-4 No	2.359	0.5314
	ApoE-4Yes	2.277	

**Table 3 jcm-10-03700-t003:** Estimated mixed linear model on the logarithmically transformed LTL (table of fixed effects).

	Estimates	Standard Error	95% Confidence Interval	T Value	*p*-Value
Intercecpt	6.9129	0.3602	6.2068; 7.6189	19.1891	<0.001
Age	−0.0796	0.0043	−0.0881; 0.0711	−18.4265	<0.001

**Table 4 jcm-10-03700-t004:** Determinants of survival. Estimated semi-parametric Cox’s proportional hazards model for survival time.

	Estimates	Standard Error	Hazard Ratio	95% Confidence Interval Limits		Z Value	*p*-Value
				Lower	Upper		
Age	0.0746	0.0077	1.0775	1.0613	1.0938	9.6721	<0.001
LTL	−0.1862	0.0486	0.8301	0.7547	0.9131	−3.8285	<0.001
Male	0.549	0.0939	1.7316	1.4405	2.0814	5.8469	<0.001
CCI	0.0921	0.0259	1.0965	1.0422	1.1536	3.5549	<0.001
SPPB	−0.1055	0.0155	0.8999	0.8729	0.9276	−6.7971	<0.001

Age in years; CCI = Charlson comorbidity index; LTL = Leukocyte telomere length; Male = Male gender; SPPB = Short Physical Performance Battery.

**Table 5 jcm-10-03700-t005:** Determinants of survival. Estimated semi-parametric Cox’s proportional hazards model for survival-time with episode splitting and weighed sample.

	Estimates	Standard Error	Hazard Ratio	95% Confidence Interval Limits		Z Value	*p*-Value
				Lower	Upper		
Age	0.0816	0.0084	1.085	1.0673	1.1030	9.7344	<0.001
LTL	−0.2538	0.0519	0.7759	0.7008	0.8589	−4.886	<0.001
Male	0.5595	0.1044	1.7499	1.4260	2.1471	5.3614	<0.001
CCI	0.1533	0.0284	1.1657	1.1026	1.2324	5.401	<0.001
SPPB	−0.0919	0.0165	0.9122	0.8832	0.9422	−5.5876	<0.001

Age in years; CCI = Charlson comorbidity index; LTL = Leukocyte telomere length; Male = Male gender; SPPB = Short Physical Performance Battery.

**Table 6 jcm-10-03700-t006:** Results from logistic regression model exploring the relationship between telomere elongation (dichotomous variable) and characteristics of the study population. Model selection based on Rao test (*n* = 613).

	Estimates	Standard Error	Odd Ratio	95% Confidence Interval Limits		Z Value	*p*-Value
				Lower	Upper		
Intercept	−2.4259	0.5663	0.0884	0.0291	0.2682	−4.2834	<0.0001
Male	−0.8237	0.3363	0.4388	0.2270	0.8483	−2.4495	0.0143
SPPB	0.1434	0.0528	1.1542	1.0407	1.2800	2.7172	0.0066
Education	0.0915	0.0336	1.0958	1.0260	1.1704	2.7237	0.0065
DCI	−0.3696	0.1361	0.6912	0.5292	0.9023	−2.7151	0.0066

DCI = Disease count index; Education in years; Male = Male gender; SPPB = Short Physical Performance Battery.

## Data Availability

The data presented in this study are obtainable on request from the corresponding author.

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
