# Peer review of "Longer Leukocytes Telomere Length Predicts a Significant Survival Advantage in the Elderly TRELONG Cohort, with Short Physical Performance Battery Score and Years of Education as Main Determinants for Telomere Elongation"

_jcm, 2021, doi:10.3390/jcm10163700_

Round 1

Reviewer 1 Report

I read the works with pleasure, the work is interesting. The authors use modern research techniques on a large population of over 600 people. I suggest removing minor editorial problems, such as, for example, additional punctuation marks on page 6 line 177. When it comes to substantive comments, I have two questions: 1. Was telomere length vs gender analysis performed? In the characteristics of the population, the authors indicate that it is a population selected from a larger group in which the life expectancy of women differs significantly from that of men. 2. Since I did not get to the work describing the entire TRELONG project, the second question is whether data on the eating habits of the respondents were collected?

Author Response

Padova, August 16, 2021

Special Issue Editors - "Translational Research in Aging, Geriatrics and Gerontology"

Dr. Joshua D. Brown

Center for Drug Evaluation & Safety, Department of Pharmaceutical Outcomes & Policy, University of Florida College of Pharmacy, Gainesville, FL 32610, USA

Dr. Robert T. Mankowski

Institute on Aging, University of Florida, Gainesville, Florida, USA

Dear Editors,

Please find enclosed herewith a copy of our revised manuscript by Pavanello et al., entitled:  Longer leukocytes telomere length predicts a significant survival advantage in the elderly TRELONG cohort, with Short Physical Performance Battery score and years of education as main determinants for telomere elongation.”

The manuscript has been carefully modified, taking into account comments and suggestions from the reviewers. We take the opportunity to thank the reviewers for giving us the opportunity to improve our work. In the revised manuscript all corrections are highlighted in yellow. The revised manuscript has been approved by all the authors.

Below, we report the responses to the specific points raised.

Looking forward to hearing from you,

Yours sincerely,

Sofia Pavanello

REVIEWER 1

Comments and Suggestions for Authors

I read the works with pleasure, the work is interesting. The authors use modern research techniques on a large population of over 600 people.

 Comment 1

I suggest removing minor editorial problems, such as, for example, additional punctuation marks on page 6 line 177.

Answer 1

From the editorial point of view, we revised all the manuscript.

When it comes to substantive comments, I have two questions:

Comment 2

Was telomere length vs gender analysis performed? In the characteristics of the population, the authors indicate that it is a population selected from a larger group in which the life expectancy of women differs significantly from that of men.

Answer 2

We evaluated the effect of gender and also the age on telomere length, as described in lines 248-250, but then in the results in line 317-325, gender was not significant for the same age. For this reason, we report only the effect of age in table 3. The fact that gender is significant (Cox's models) for predicting survival (and therefore that there is a significant difference in life expectancy between men and women) does not imply that our gender should be significant in predicting telomere length.

Comment 3

Since I did not get to the work describing the entire TRELONG project, the second question is whether data on the eating habits of the respondents were collected?

Answer 3

Data on eating habits were collected in 2003 (baseline) by questioning the participants about the regular intake (yes/no) of a list of foods. The statistical analysis showed that answers were substantially the same for the whole sample, suggesting all participants had the same eating habits. This specification was added to the line 114-118 of the revised manuscript.

Reviewer 2 Report

In this manuscript, Pavanello et al. describe results from a longitudinal study of telomere length associations with various metrics of human health. This is an extremely valuable study and the data are very important to appreciate. Some trends are not surprising, but helpful as they serve as sort of internal controls for the study’s data quality. Other results appear more novel, and again the impact of the longitudinal data on the telomere and aging fields are significant. Overall, I think the work is sound and important, thought he writing and displaying of the results needs some improvement for clarity. See Major and Minor concerns below.

Major:

Have the authors, or others, benchmarked the real-time PCR method used here for telomere length determination against other established methods for quantifying how long telomeres are, such as Southern blotting, etc.? It seems from the referencing of other papers that this method is invented by this manuscript’s authors, but please make this clear in the manuscript is this is not the case.

Table 2 and its description in the Results is not possible for me to follow, perhaps since I am not a statistician. What was the point of this table and can that be simply stated at the beginning of the paragraph on line 348? As written, this paragraph starts off with just talking about an esoteric statistical test name; the authors need to boil down the jargon here, and elsewhere in the manuscript, to explicitly state the findings in “plain language.” Lines 355–363 are good.

Table 2 columns are not defined; the last one is presumably the P value… these columns need titles, especially since there is no legend for tables in the journal, presumably.

T/S is never defined, yet is a critical unit throughout the manuscript. As a basic telomere researcher, I don’t even know what it stands for. It must be defined at least in the Introduction and Figure Legends.

The current supplementary figures and tables should be part of the  main manuscript; they are not optional reading in my opinion, but rather are central to the data being reported.

Too much reliance on acronyms and esoteric jargon in the abstract, and elsewhere, for the non-specialist reader to understand the gist of the manuscript.

English/grammar need some attention for professional publishing (incorrect prepositions in certain phrases, which vs. that errors, many commas missing).

I do not understand what the authors mean in the critical first sentence of the Discussion with use of the word “monotonous.” It should be rephrased.

Around line 439, the authors describe the “paradoxical” increase in telomere length in some study participants, and suggest it may mean telomerase has become more active in these individuals’ leukocytes. This would suggest to me they might also have a great chance of cancer, since telomerase can be oncogenic. Did the authors look at this possible correlation in the data?

Line 493: there is something grammatically wrong with the sentence that starts on this line, since it does not make sense… maybe a word missing? Also, I find a flaw in the logic: less immigration should mean a more inbred population, and thus a greater chance for genetic problems, not less. The authors should address this.

Minor:

The first paragraph of the Discussion is hard to deal with since it is so long; I recommend that the authors break it up a bit to make it easier to appreciate.

Line 305: is “tendential” the right word? I don’t know it

Line 482: “surprisingly” should be “surprising”

Author Response

Padova, August 16, 2021

Special Issue Editors - "Translational Research in Aging, Geriatrics and Gerontology"

Dr. Joshua D. Brown

Center for Drug Evaluation & Safety, Department of Pharmaceutical Outcomes & Policy, University of Florida College of Pharmacy, Gainesville, FL 32610, USA

Dr. Robert T. Mankowski

Institute on Aging, University of Florida, Gainesville, Florida, USA

Dear Editors,

Please find enclosed herewith a copy of our revised manuscript by Pavanello et al., entitled:  Longer leukocytes telomere length predicts a significant survival advantage in the elderly TRELONG cohort, with Short Physical Performance Battery score and years of education as main determinants for telomere elongation.”

The manuscript has been carefully modified, taking into account comments and suggestions from the reviewers. We take the opportunity to thank the reviewers for giving us the opportunity to improve our work. In the revised manuscript all corrections are highlighted in yellow. The revised manuscript has been approved by all the authors.

Below, we report the responses to the specific points raised.

Looking forward to hearing from you,

Yours sincerely,

Sofia Pavanello

REVIEWER 2

Comments and Suggestions for Authors

In this manuscript, Pavanello et al. describe results from a longitudinal study of telomere length associations with various metrics of human health. This is an extremely valuable study and the data are very important to appreciate. Some trends are not surprising, but helpful as they serve as sort of internal controls for the study’s data quality. Other results appear more novel, and again the impact of the longitudinal data on the telomere and aging fields are significant. Overall, I think the work is sound and important, thought he writing and displaying of the results needs some improvement for clarity. See Major and Minor concerns below.

Major:

Comment 1

Have the authors, or others, benchmarked the real-time PCR method used here for telomere length determination against other established methods for quantifying how long telomeres are, such as Southern blotting, etc.? It seems from the referencing of other papers that this method is invented by this manuscript’s authors, but please make this clear in the manuscript is this is not the case.

Answer 1

We used the quantitative Real-Time PCR technique to assess LTL in triplicate. This is the most frequently applied method for LTL testing and, as reported in the paragraph of “Strengths and limitations”, this fast and highly sensitive method requires relatively small amounts of DNA and allows high-throughput. However, the small amounts of DNA available did not allow us to perform further analyzes with others existing methods such as Southern blotting

Furthermore, by using an almost totally automated workflow from DNA extraction to LTL analysis, pre-analytical factors and analytical factors did not differ among analyses and the same pool DNA was used as a reference standard in each q-RT PCR reaction. In this way the interbach variation was under control, minimizing measurement error and warranting a coefficient of variation (CV) less than 9%.

As suggested by Reviewer 2, we have also modified the sentence concerning the LTL analysis by qPCR, replacing the term “developed” with “described” and including “according to the reference method [Cawthon, 2009 reference 28]” at lines 211 of the revised manuscript.

Comment 2

Table 2 and its description in the Results is not possible for me to follow, perhaps since I am not a statistician. What was the point of this table and can that be simply stated at the beginning of the paragraph on line 348? As written, this paragraph starts off with just talking about an esoteric statistical test name; the authors need to boil down the jargon here, and elsewhere in the manuscript, to explicitly state the findings in “plain language.” Lines 355–363 are good.

Answer 2

Table 2 becomes Table 4 in the revised manuscript, its title, and its description have been reformulated at lines 359-363 and 376-378 of the revised manuscript. The point of this table (and of the adapted model) is to show that telomere length has a significant effect on survival even in a multivariate analysis which also takes into account the effect of other factors such as age, gender, the CCI and the SPPB.

The statement “Since the analysis of the Schoenfeld residuals gave a non-significant result (p= 0.92) for each covariate and for the model as a whole, the hypothesis of proportionality of the risks was accepted.” was removed; it now appears in paragraph 2.9.3 of Statistical Analysis after “… whole.”

These specifications have been added to the revised manuscript at lines 267-269.

Comment 3

3) Table 2 columns are not defined; the last one is presumably the P value… these columns need titles, especially since there is no legend for tables in the journal, presumably.

Answer 3

In table 2 all columns present titles, however they are not clearly visible because they are displayed in the previous page with respect to the data. In the revised manuscript Table 2 becomes Table 4 and now is completely shown at page 14.

Comment 4

T/S is never defined, yet is a critical unit throughout the manuscript. As a basic telomere researcher, I don’t even know what it stands for. It must be defined at least in the Introduction and Figure Legends.

Answer 4

T/S ratio is the unit of measurement of LTL that is the proportion of telomere repeat copy number (T) in relation to a single copy gene (S) (T/S ratio). This further information has been better specified in paragraph 2.8 at line 214 of the revised manuscript.

 Comment 5

The current supplementary figures and tables should be part of the main manuscript; they are not optional reading in my opinion, but rather are central to the data being reported.

Answer 5

As suggested by Reviewer 2 all supplementary figures and tables have been included in the main text of the revised manuscript. We have therefore re-numbering all the figures and tables in the revised manuscript. Now there are in total: 6 tables and 4 figures.

 Comment 6

Too much reliance on acronyms and esoteric jargon in the abstract, and elsewhere, for the non-specialist reader to understand the gist of the manuscript.

Answer 6

In accordance with the reviewer's suggestions we have modified the abstract to make understanding easier for the non-specialist reader.

 Comment 7

English/grammar needs some attention for professional publishing (incorrect prepositions in certain phrases, which vs. that errors, many commas missing).

Answer 7

The English writing was revised through all the revised manuscripts.

Comment 8

I do not understand what the authors mean in the critical first sentence of the Discussion with the use of the word “monotonous.” It should be rephrased.

Answer 8

Monotonic in Mathematics is a function or quantity varying in such a way that it either never decreases or never increases, namely, a steady relationship. This specification has been added to the revised manuscript at line 404. In our study, there is a steadily increasing relationship between telomere length and survival, and this can be seen from the fact that survival curves and median survival times respect the ordering of telomere length quartiles.

Furthermore, the statement at the begging of the discussion “The key findings stemming from our study are an increasing monotonous dependence between LTL and survival: the longer the LTL, the longer survival and, therefore, a lower mortality risk is estimated.” was rephrased as follows. “The key findings from our study show a steadily increasing relationship between LTL and survival: the longer the LTL, the greater the survival and, therefore, a lower mortality risk is estimated.”

Comment 9

Around line 439, the authors describe the “paradoxical” increase in telomere length in some study participants and suggest it may mean telomerase has become more active in these individuals’ leukocytes. This would suggest to me they might also have a great chance of cancer, since telomerase can be oncogenic. Did the authors look at this possible correlation in the data?

Answer 9

We used the term “paradoxical” to describe the increase in LTL after follow-up as it is unusual and difficult to understand with respect to the age-dependent LTL shortening. However, there may be exceptions. The fact that telomerase has become more active in these individuals could be excluded since cancer, which was considered among the chronic diseases, was not significantly related to LTL. Furthermore, cancer also included in the Charlson comorbidity index (CCI) and Disease count index (DCI). However, only the general DCI was found to be significant but negatively correlated with telomere elongation, suggesting that telomerase might be not activated.

Comment 10

Line 493: there is something grammatically wrong with the sentence that starts on this line, since it does not make sense… maybe a word missing? Also, I find a flaw in the logic: less immigration should mean a more inbred population, and thus a greater chance for genetic problems, not less. The authors should address this.

Answer 10

We agree with Reviewer 2, sentences have been modified in the revised manuscript at lines 519-522 as follows. In the ancestors of Northern Italian climate-related selective pressures has influenced an adaptive evolution that make Northern Italy less prone to develop accelerating aging disorders, including diabetes type 2 and obesity [66].

Minor:

Comment 11

The first paragraph of the Discussion is hard to deal with since it is so long; I recommend that the authors break it up a bit to make it easier to appreciate.

Answer 11

We have modified the paragraph according to the suggestion of the reviewer and now Discussion was divided into 5 paragraphs.

 Comment 12

Line 305: is “tendential” the right word? I don’t know it

Answer 12

The term indicates a tendency in reduction, the term however was deleted in the revised manuscript.

 Comment 13

Line 482: “surprisingly” should be “surprising”

Answer 13

The term “surprisingly” was replaced with “surprising” at line  505  of the revised manuscript.
